# Association of Dietary Protein Intake with Muscle Mass in Elderly Chinese: A Cross-Sectional Study

**DOI:** 10.3390/nu14235130

**Published:** 2022-12-02

**Authors:** Yifei Ouyang, Feifei Huang, Xiaofan Zhang, Li Li, Bing Zhang, Zhihong Wang, Huijun Wang

**Affiliations:** Key Laboratory of Trace Element Nutrition of National Health Commission, National Institute for Nutrition and Health, Chinese Center for Disease Control and Prevention, Beijing 100050, China

**Keywords:** dietary protein, muscle mass, aging

## Abstract

Most data regarding the association between dietary protein intake and muscle mass come from developed Western countries. This cross-sectional study investigates the association between the amount and distribution of dietary protein intake and muscle mass in elderly Chinese adults. This analysis includes 4826 participants aged 60 years and above from the dataset of the China Health and Nutrition Survey (CHNS) 2018. Protein intake data were assessed using 3-day, 24 h dietary recalls. Appendicular skeletal muscle mass (ASM) was obtained using the bioelectrical impedance analysis (BIA). Two-thirds of dietary protein intake comes from plant sources in the elderly Chinese population. The median total dietary protein intake was 60.5 g/d in low muscle mass participants for males and 52.5 g/d for females, which was lower than for their respective counterparts. Compared to the lowest quartile of protein intake, the highest total protein intake group had increased muscle mass by 0.96 kg among men and by 0.48 kg among women (*p* < 0.0001), and the highest vegetable protein intake group had increased muscle mass by 0.76 kg among men and by 0.35 kg among women (*p* < 0.0001). The amount of dietary protein intake with each meal was less than 20 g. High total protein intake and high plant-based protein intake were positively associated with higher muscle mass. A U-shape was observed between total dietary protein intake and low muscle mass risk in elderly Chinese adults. It should be encouraged to increase total daily protein intake to maintain muscle health.

## 1. Introduction

China is experiencing serious challenges due to an appreciable increase in its elderly population [1]. The size of the elderly Chinese population aged 60 or older increased to 264 million, which was approximately 18.7% of the total population at the time, in 2020 [2]. Aging-induced endocrine alterations influence the related decline in bodily function [3], including skeletal muscle mass and function. Muscle mass declines after peaking in the fourth decade and may result in sarcopenia when muscle mass loss is more than 30% of the body weight [4]. Sarcopenia is a loss of skeletal muscle mass with subsequent low muscle strength, leading to serious health consequences, increased fall and fracture risk, frailty, diminished quality of life, and even increased mortality risk [5]. Therefore, sarcopenia is recognized as an independent condition by the International Classification of Disease, Tenth Revision, Clinical Modification (ICD-10-CM) Code [6]. Research interest in sarcopenia has recently burgeoned internationally.

As of now, nutrition, exercise, and pharmacologic agents might be the three important interventions to improve muscle mass, strength, and function [7]. Studies have demonstrated that individuals with higher protein intake have a higher muscle mass [8,9], which is a protective factor against sarcopenia. The adequate consumption of dietary protein is critical for the maintenance of muscle mass. The majority of the population of Western countries has a relatively high protein intake and exceeds minimum recommendations for protein intake compared to other countries [10,11]. However, our previous study showed that there is a high prevalence of inadequate protein intake in the elderly Chinese population [12]. More importantly, recent evidence pointed to current recommendations being insufficient to meet the minimum protein requirement to counteract muscle loss in older adults [13,14,15]. In fact, relatively few studies have examined issues, such as how much and what type of dietary protein may provide the greatest value for muscle mass among the elderly Chinese population. Apart from the total dietary protein intake, there is evidence that nitrogen turnover is not only influenced by the amount, but also by protein intake per meal [16]. Previous reports showed that an even distribution of daily protein intake was associated with increased muscle mass [17,18]. It is suggested that the consumption of about three meals a day, each containing about 30 g of protein, is optimal for the stimulation of 24-h muscle protein synthesis, both for younger and healthy older adults [19,20]. However, the distribution of dietary protein intake among the elderly Chinese population remains unknown.

This study, an update of our previous work [12], characterizes different dietary intake between normal muscle mass and low muscle mass, and investigates the association between the amount and different sources of dietary protein intake and muscle mass in elderly Chinese adults. Importantly, the current study evaluates the distribution of dietary protein in older Chinese adults, providing the latest evidence on another important factor associated with muscle mass: aging.

## 2. Materials and Methods

### 2.1. Data Source

Data used in this cross-sectional study were from the CHNS. The CHNS was a longitudinal household-based study of the Chinese population conducted in a collaboration between the University of North Carolina at Chapel Hill and the National Institute for Nutrition and Health, Chinese Center for Disease Control [21]. The CHNS used a stratified multistage sampling method to minimize the selection bias and aimed to provide a wide range of socio-demographic characteristics, diet, health, physical activity, behaviors, and environmental changes to measure the effects on the health of the Chinese population. In the latest 2018 survey, the CHNS added the BIA technique to assess body composition. After the exclusion of disabled persons and those with missing data for key variables, such as age, gender, diet, and body composition, a total of 4826 subjects aged 60 years and above was selected to participate in this study. The final protocol of the CHNS was approved by the institutional review committees of the National Institute for Nutrition and Health, Chinese Center for Disease Control and Prevention and by the University of North Carolina at Chapel Hill (201524). Informed consent was obtained from all subjects before participating in the study.

### 2.2. Dietary Assessment

Personal dietary data were obtained using 3-day 24-h dietary recalls (two weekdays and one weekend day) through face-to-face interviews. We collected detailed information of all foods and beverage consumed by using a tablet (ThinkPad Tablet2, Lenovo, Beijing, China). We also used food models and pictures, in addition to food weighing, to improve the accuracy of recalls. Dietary carbohydrate intake (g), fat intake (g) and dietary protein intake (g) from each food the participants consumed were calculated according to the China Food Composition Table. The total energy intake was computed as the sum of energy from proteins, fats, and carbohydrates. We computed the total energy intake by summing up energy from carbohydrates, fats and proteins.

We defined plant source protein as that stemming from cereals, tubers, starches, legumes, vegetables, fruits, fungi, algae, nuts, and seeds. Animal source protein was defined as stemming from meat, poultry, milk, eggs, and aquatic products. Dietary protein intake was expressed two ways: (1) grams of protein per day (g/d). (2) grams of protein per kilogram of actual body weight per day (g/kg BW/d). We computed the amount of protein ingested in each meal by summing up the amounts of protein of the all protein sources per meal [16].

The distribution of protein intake for each participant over the day was estimated using a coefficient of variation (CV = SD/mean value) of protein intake (g/meal) from breakfast, lunch, and dinner [18,22]. Then, we calculated the mean CV averaged over the three days. The more even the distribution is, the lower the protein CV values are.

### 2.3. Outcome and Covariate Measurements

Muscle mass was measured by the BIA (TANIATA, BC-601, Tokyo, Japan). We calculated ASM by summing up skeletal muscle in the arms and legs. The skeletal muscle mass index (SMI) = ASM (kg)/height (meter)^2^. Low skeletal muscle mass was diagnosed according to the AWGS 2019 recommendation (Male: SMI < 7.0 kg/m^2^, Female: <5.7 kg/m^2^) [5]. The body mass index (BMI) was calculated as the weight in kilograms divided by height in meters squared. Height was measured with participants standing without shoes to the nearest 0.1 cm using a stadiometer (SECA 206 Hamburg, Germany). Weight in light clothes was also measured without shoes to the nearest 0.1 kg using a calibrated beam scale (TANIATA, BC-601, Tokyo, Japan). Being overweight and obesity were defined based on BMI cut-off points of 24 and 28, based on the Chinese definitions.

Questionnaires were applied during in-person interviews to collect individuals’ information. Participants were grouped into several groups: gender (male and female), age (60–69 years and 70 years and above), education (primary/illiterate and middle school and above), and residential areas (city and village).

Physical activity was measured as the average metabolic equivalents of task (MET) hours per week, including four PA domains: occupational, household, travel, and leisure time activities. We computed the average MET-hours per week by multiplying the average intensity of each activity and the time spent in each activity. Sedentary leisure time was calculated as the average hours per week. The calculation of total physical activity and leisure sedentary time has been described in previous studies [23,24].

Venous blood samples were collected in the morning after an overnight fast by a trained nurse and an array of biochemical indexes were measured in a national lab in Beijing with strict quality control. According to diagnostic criteria recommended by the World Health Organization in 1999 [25], type 2 diabetes mellitus (T2DM) was diagnosed as fasting plasma glucose (FPG) ≥ 7.0 mmol/L or HbA1C ≥ 6.5, self-reported history of diabetes diagnoses, or use of antidiabetic medication. Impaired glucose regulation (IGR) was defined as FPG ≥ 6.1 and <7.0 mmol/L. An FPG concentration < 6.1 mmol/L was considered normal glucose tolerance (NGT).

### 2.4. Statistical Analysis

Continuous variables were expressed as medians (interquartile range) and compared by Kruskal-Wallis tests. Categorical variables were expressed as numbers (percentages) and compared by Chi-square tests. Multiple linear regression models were used to analyze the association between total dietary protein, plant-based protein, animal-based protein, and muscle mass. Protein intake was evaluated as a continuous variable by using sex-specific quartiles. Linear trends tests across the quartiles of protein intake were conducted by using the median value of each protein category as a continuous variable in the linear regression models.

To test the dose-response association between total dietary protein intake and low muscle mass outcomes, dietary protein intake was entered in the logistic regression model as a restricted cubic spline with 3 knots (located at the 5th, 50th and 95th). The zero value of total dietary protein intake was chosen as the reference group for all spline plots. We adjusted all covariates into models, such as age, gender, education, residence area, total energy intake, BMI, total physical activity, leisure sedentary time, and diabetes. All data analyses were performed by SAS 9.4 (SAS Inc., Cary, NC, USA). Significance level was set at *p* value < 0.05.

## 3. Results

### 3.1. Participant Characteristics

As shown in Table 1, a total of 4826 elderly adults participated in the present study. The prevalence of low muscle mass in the study population was 9.8%. There were significant differences between the normal muscle mass group and low muscle mass group in terms of age, gender, education level, and residence area. The normal muscle mass group was more likely to have a higher total energy intake, BMI, muscle mass, relative skeletal muscle index, and physical activity.

Table 2 shows the amount of daily dietary protein intake and that from animal source protein and plant source protein. The median total dietary protein intake was 60.5 g/d in low muscle mass participants for males and 52.5 g/d for females, which was lower than that of their respective counterparts. The plant protein intake among males in the normal muscle mass group was 31.8 g/d and 27.9 g/d for females, which was higher than that of their respective counterparts. We found no significant differences in the animal source protein between the normal muscle mass and the low muscle mass groups in males or females.

### 3.2. Association between Dietary Protein Quantity, Quality, and Muscle Mass

Table 3 shows the association between dietary protein quantity, quality and muscle mass. After adjusting for all covariates, higher levels of total dietary protein intake and vegetable protein intake were associated with higher muscle mass, showing an increasing trend (*p* _trend_ < 0.05). Compared with the reference group (the lowest total protein intake group), the highest total protein intake group had increased muscle mass by 0.96 kg among men and by 0.48 kg among women. Similarly, the highest vegetable protein intake group had increased muscle mass by 0.76 kg among men and by 0.35 kg among women compared to the reference group (the lowest vegetable protein intake group). However, the association between animal protein and muscle mass was not significant.

As shown in Figure 1 and Figure 2, the dose-response analyses of restricted cubic spline logistic regression showed a significant non-linear association of total dietary protein intake with risk of low muscle mass in both men and women (*p* < 0.05). In the spline regression model, compared with subjects who had no dietary protein intake, the subjects with a 78.0 g/day protein intake had a decreased risk of low muscle mass by 62% (*OR* = 0.38, 95% CI: 0.17–0.87, *p* < 0.001) among males. When compared with subjects who had no dietary protein intake, the subjects with a 68.0 g/day protein intake had a decreased risk of low muscle mass by 66% (*OR* = 0.34, 95% CI: 0.15–0.77, *p* < 0.001) among females.

### 3.3. Association between Dietary Protein Distribution and Muscle Mass

As shown in Table 4, the percentage of protein intake in the morning was 25%, which was lower than that for intake at noon (36%) and in the evening (36%) among the men in the normal muscle mass group. No significant difference was found between the normal muscle mass group and the low muscle mass group in either men or women (*p* > 0.05). Similarly, the amount of dietary protein intake in each meal was less than 20 g, and no significant difference was found across the three daily mealtimes in different groups. Figure 3 shows that the CV of protein distribution was not significant between the normal muscle mass group and the low muscle mass group in both men and women (*p* > 0.05).

## 4. Discussion

In this study, we found that the plant-based protein was the major source of dietary protein in elderly Chinese adults. There is a significant positive association between the amount of total dietary protein intake and vegetable protein intake between normal muscle mass participants and low muscle mass participants among both men and women. Although the daily protein intake distribution was equal across breakfast, lunch, and dinner, the intake of protein per meal was less than 20 g. There are some elderly Chinese individuals potentially vulnerable to the risk of low muscle mass, especially those of older ages, those with primary education/illiteracy, those living in village regions, and males.

We observed that the amount of total dietary protein intake and plant protein intake in normal muscle mass participants were higher than in their respective counterparts. These results are in line with a previous study showing the difference between sarcopenia subjects and non-sarcopenia subjects [26]. In the current study, we found an association between high plant protein intake and muscle mass among the elderly Chinese population, but animal protein was not associated with muscle mass. There might be explanations for this. First, the traditional Chinese type of diet is charactered by large amounts of cereals and vegetables. Thus, plant protein intake contributed more to the total dietary protein intake than animal protein intake. It is possible that the ingestion of greater amounts of vegetable-source proteins may be achieve the same anabolic response evoked by smaller quantities of animal-source proteins [27,28]. Although animal-based protein has been widely acknowledged to have better digestibility and bioavailability than plant-based protein, the ascorbic acid found in vegetables and fruits can enhance plant protein absorption, not to mention greater amounts of plant-based proteins are consumed per meal. Second, the intake of animal-source food was smaller than that of vegetable-source food in elderly Chinese individuals, which may have limited our ability to detect a significant association between animal-based protein and muscle mass. Conversely, the Western dietary pattern is well-known to be a predominantly animal-based diet, which indicated a greater animal protein intake. Epidemiological studies from developed countries have documented that animal protein was significantly associated with changes in lean muscle, but this was not the case for vegetable protein in older adults [29]. Although protein from animal sources is a source of high biological value that can provide all essential amino acids, it is also rich in saturated fatty acids, cholesterol, and calories. Thus, in order to enhance the nutritional quality, the Chinese Nutrition Society suggests people to consume cereals and legumes together.

Indeed, the Chinese diet has experienced a shift towards more animal-based diets and fewer plant-based diets, including in the elderly population. The gap between plant protein intake and animal protein intake might become smaller [12]. Moreover, the China Nutrition and Health Surveillance (2010–2013) reported that the proportion of protein intake from cereals and potatoes was higher in the elderly living in the village than in those living in the city, and the proportion of protein intake from livestock, poultry, and aquatic products was higher in the elderly living in the city than in those living in the village. Our CHNS research group will follow this trend and identify the association between animal protein and mass muscle and sarcopenia in future analyses.

Furthermore, there is evidence for the positive association of high dietary protein intake with muscle mass [29,30], which we could also identify in our analyses. In the present study population, we found that the total dietary protein intake was positively associated with muscle mass in both men and women. At present, several organizations, such as the International Working Group on Sarcopenia, the European Working Group on Sarcopenia in Older People, and the Asian Working Group on Sarcopenia, have suggested the importance of adequate protein intake for preventing low muscle mass or sarcopenia without the optimal level of recommended protein intake. By contrast, the ESPEN Expert Group and the PROT-AGE Study Group recommended an average daily intake of at least 1.0–1.2 g protein/kg body weight for healthy older people to maintain and regain lean body mass and function [31,32]. Our study is the first to investigate the association between dietary intake and muscle mass among the elderly Chinese population. We found that there is a near U-shaped dose-effect relationship of total dietary intake and risk of low muscle mass. Compared with subjects who had no dietary protein intake, a dietary protein intake above 78.0 g/d in males and 68.0 g/d in females might be a threshold point for preventing low muscle mass in elderly Chinese adults as indicated by our study. These results are greater than the current Estimated Average Requirements (EAR) and the recommended nutrient intake (RNI) for older adults in Chinese DRIs (2013), which is 60 g/d for males and 50 g/d for females in EAR and 65 g/d for males and 55 g/d for females in RNI. As is well known, the EAR and RNI are defined as the minimum amount of dietary protein required to establish nitrogen balance, but not to prevent muscle mass loss. Meanwhile, a growing number of studies has questioned the adequacy of current protein recommendations for older adults in maintaining muscle mass and strength. Since elderly Chinese individuals are at high risk for insufficient protein intake, it is time to consider updating the protein intake recommendation to confer muscle health. Future research is thus warranted to verify the optimal level of dietary protein intake recommended in our study.

Apart from the total daily intake and the quality of consumed protein, per-meal quantity and protein distribution have also been shown to play an important role in preserving muscle mass and function. Our study observed that there is no significant difference in the percentage of protein intake in the morning, at noon, and in the evening between the normal muscle mass group and the low muscle mass group for both men and women. However, the total dietary protein intake was barely 20 g per meal, which is the minimum needed to stimulate muscle protein synthesis [33]. Previous studies have shown that the ingestion of approximately 25–40 g of protein per meal should be suggested for the maximal stimulation of muscle synthesis in older individuals [17,34]. However, according to the China National Nutrition Surveys, the consumption of eggs, total dairy products, and soybeans was only 23.5 g/d, 23.1 g/d, and 9.8 g/d [35], which was far below the recommendation from the Chinese Society of Nutrition.

The strongest advantage of this study is investigating the optimal level and distribution of protein intake in Chinese older adults by using the most recent dietary data from 15 provinces in China. One limitation of the study is its cross-sectional design, which does not allow any statement on causal relationship. Low muscle mass may plausibly affect dietary protein intake as well. Another limitation is that food consumption information using three consecutive days of 24-h dietary recalls were generally collected from summer to autumn, so seasonal variations in protein intake could not be taken into account. In addition, there is always recall bias in self-reported measurements.

## 5. Conclusions

In summary, the majority of dietary protein intake comes from plant sources in the elderly Chinese population. A high total protein intake and high plant-based protein intake were positively associated with muscle mass. A U-shape was observed between total dietary protein intake and low muscle mass risk in elderly Chinese adults. Although there is an even distribution of protein intake over the day among elderly Chinese people, the dietary protein intake was less than 20 g per meal. The Chinese elderly population should be encouraged to consume a moderate amount of protein to solicit a positive muscle synthesis response.

## Figures and Tables

**Figure 1 nutrients-14-05130-f001:**
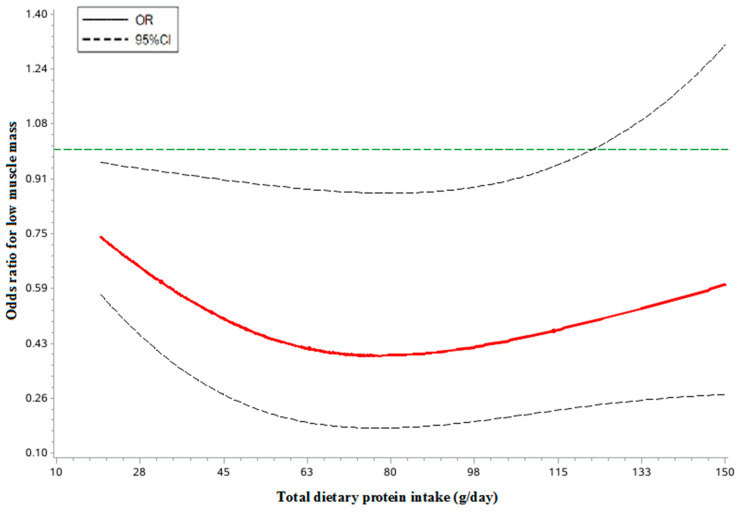
The restricted cubic spline for the association between dietary protein intake and risk of low muscle mass among males.

**Figure 2 nutrients-14-05130-f002:**
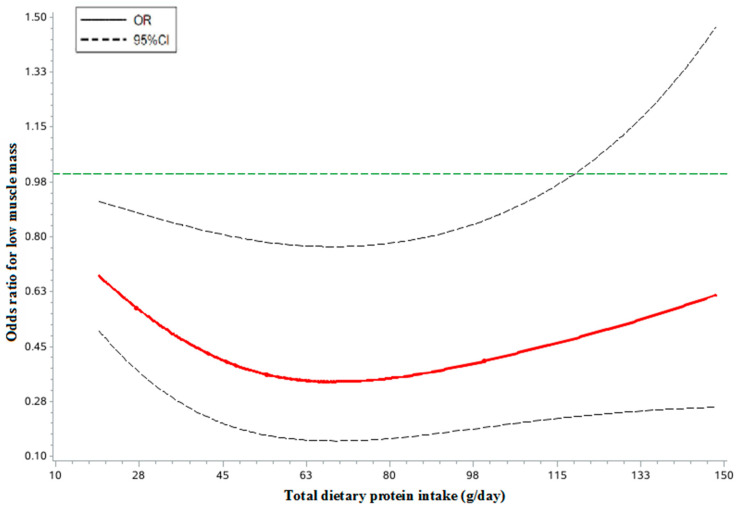
The restricted cubic spline for the association between dietary protein intake and risk of low muscle mass among females.

**Figure 3 nutrients-14-05130-f003:**
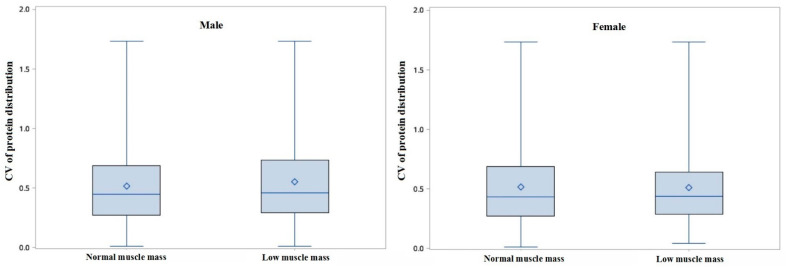
Boxplots of coefficient of variation (CV) of protein distribution over the three daily mealtimes in different groups (*p* > 0.05).

**Table 1 nutrients-14-05130-t001:** Demographic characteristics of participants.

Wave	Normal Muscle Mass	Low Muscle Mass	*p* Value
Sample size (*N*) ^b^	4350 (90.2)	476 (9.8)	
Age group (years) ^b^			
60–69	2819 (91.3)	270 (8.7)	0.0005
70–	1531 (88.1)	206 (11.9)	
Gender ^b^			
Male	1979 (88.5)	258 (11.5)	0.0003
Female	2371 (91.6)	218 (8.4)	
Education level ^b^			
Primary/illiterate	2210 (89.3)	265 (10.7)	0.0404
Middle school and above	2137 (91.1)	210 (8.9)	
Diabetes mellitus, *n* (%) ^b^			
NGT	2932 (67.5)	356 (75.1)	0.0026
IGR	889 (20.5)	79 (16.7)	
T2DM	521 (12.0)	39 (8.2)	
Residence area, *n* (%) ^b^			
City	1737 (91.9)	153 (8.1)	0.0009
Village	2613 (89.0)	323 (11.0)	
Total energy intake, kcal/d ^a^	1817.0 (819.1)	1755.1 (867.3)	0.0245
Body mass index, kg/m^2 a^	24.7 (4.4)	19.8 (2.8)	<0.0001
Muscle mass, kg ^a^	18.0 (7.3)	15.0 (4.2)	<0.0001
Relative skeletal muscle index, kg/m^2 a^	7.2 (1.8)	5.9 (1.1)	<0.0001
Total physical activity, MET-h/w ^a^	68.3 (105.0)	60.7 (116.9)	0.0350
Leisure sedentary time, h/w ^a^	16.0 (16.0)	17.5 (16.0)	0.0226

^a^ Continuous values are expressed as median (interquartile range) and compared by Kruskal-Wallis tests. ^b^ Categorical values are expressed as numbers (percentages) and compared by chi square tests.

**Table 2 nutrients-14-05130-t002:** The amount of daily dietary protein intake and food sources of protein in different groups ^a^.

Dietary Protein Intake	Male			Female		
	Normal Muscle Mass	Low Muscle Mass	*p* Value	Normal Muscle Mass	Low Muscle Mass	*p* Value
Total protein (g/d)	63.2 (32.0)	60.5 (34.9)	0.0491	54.4 (28.4)	52.5 (35.0)	0.0426
Animal protein (g/d)	22.0 (22.5)	22.7 (24.9)	0.3128	18.6 (18.6)	18.3 (18.5)	0.4481
Plant protein (g/d)	31.8 (18.8)	30.2 (17.4)	0.0477	27.9 (16.9)	26.1 (15.6)	0.0468

^a^ The values are expressed as median (interquartile range) and compared by Kruskal-Wallis tests.

**Table 3 nutrients-14-05130-t003:** Association between dietary protein intake and muscle mass ^a^.

Variables	Male				Female			
	Quartiles of Protein Intake	Coefficient	*SE*	*p* Value	Quartiles of Protein Intake	Coefficient	*SE*	*p* Value
Total protein intake (g/d)								
Quartile 1 (lowest)	<48.0	Ref			<41.3	Ref		
Quartile 2	48.0–62.8	0.52	0.25	0.0373	41.3–54.2	0.30	0.11	0.0059
Quartile 3	62.8–80.5	0.68	0.24	0.0049	54.2–70.0	0.47	0.11	<0.0001
Quartile 4 (highest)	>80.5	0.96	0.24	<0.0001	>70.0	0.48	0.12	<0.0001
*p* value for trend				<0.0001				<0.0001
Animal protein (g/d)								
Quartile 1 (lowest)	<12.3	Ref			<10.8	Ref		
Quartile 2	12.3–22.1	−0.12	0.23	0.6136	10.8–18.5	−0.10	0.11	0.3476
Quartile 3	22.1–35.1	0.23	0.22	0.2926	18.5–29.3	0.11	0.11	0.3315
Quartile 4 (highest)	>35.1	−0.13	0.21	0.5557	>29.3	0.10	0.12	0.4122
*p* value for trend				0.8362				0.0896
Vegetable protein (g/d)								
Quartile 1 (lowest)	<23.1	Ref			<20.1	Ref		
Quartile 2	23.1–31.5	0.35	0.24	0.0350	20.1–27.8	0.24	0.11	0.0274
Quartile 3	31.5–41.9	0.54	0.23	0.0199	27.8–36.9	0.31	0.11	0.0068
Quartile 4 (highest)	>41.9	0.76	0.22	0.0007	>36.9	0.35	0.12	0.0029
*p* value for trend				0.0006				<0.0001

Abbreviation: SE, stand error. ^a^ Model is adjusted for all covariates.

**Table 4 nutrients-14-05130-t004:** Dietary protein intake in the morning, at noon, and in the evening in different groups ^a^.

	Male			Female		
	Normal Muscle Mass	Low Muscle Mass	*p* Value	Normal Muscle Mass	Low Muscle Mass	*p* Value
Percentage of protein intake						
% morning	0.25 (0.18)	0.26 (0.19)	0.6533	0.25 (0.18)	0.25 (0.18)	0.4728
% noon	0.36 (0.18)	0.37 (0.20)	0.7106	0.37 (0.18)	0.36 (0.22)	0.5039
% evening	0.36 (0.18)	0.37 (0.19)	0.3291	0.35 (0.18)	0.37 (0.15)	0.1439
Dietary protein intake						
Morning (g/meal)	13.4 (9.7)	13.0 (8.9)	0.0986	11.9 (8.6)	11.2 (8.2)	0.4164
Noon (g/meal)	19.7 (15.8)	19.4 (20.9)	0.3401	17.6 (14.2)	17.0 (16.9)	0.3978
Evening (g/meal)	19.4 (15.9)	19.9 (17.9)	0.6941	16.8 (14.6)	16.3 (12.9)	0.9234

^a^ The values are expressed as median (interquartile range) and compared by Kruskal-Wallis tests.

## Data Availability

Data sharing is not applicable to this article.

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
