# Peer review of "Association of Dietary Protein Intake with Muscle Mass in Elderly Chinese: A Cross-Sectional Study"

_nutrients, 2022, doi:10.3390/nu14235130_

Round 1

Reviewer 1 Report

This study showed the significant association between protein intake (amount, source, and distribution) and muscle mass in elderly Chinese and suggests that increased protein intake is necessary to maintain muscle health.  Overall, this manuscript is well-organized and well-thought-out. However, I recommend that the authors address the following comments before publishing this article.

1. Define protein intake and low muscle mass.

·      What are the cutoff protein intake levels for Q 1-4?

·      What were the cutoff low muscle mass index levels for the male and female participants for the definition of low muscle mass?

2. What solid and dashed lines are indicated in figures 1 and 2? Label them in the figures.

3. Provide a citation for the sentence in lines 33-34.

4. English language needs to improve significantly. I suggest that authors have someone proficient in English revise the manuscript. Below are a few examples:

·      Line 26: muscle mass

·      Line 27: should be encouraged

·      Line 28: maintain

·      Line 37: result in

·      Line 44: exercise

·      Line 48: population

·      Line 51: what is the EAR, RNI? (full-form)

·      Line 73: conducted in collaboration

·      Line 76: minimize

·      Line 80: composition was

·      Line 110: ASM (KG)/height2    
                Single space between the word and parenthesis

    Units for height (meters?)

·      Line 137: Correct the sign HbA1C

·      Line 151: a restricted cubic spline

·      Line 178: instead of saying no difference, I would say no significant difference

·      Define β in table 3

·      Table 4 caption: in the evening

·      Line 240: delete another

·      Line 290: body weight

·      Line 304: stimulate

Author Response

please check out the upload replay, thanks.

Reviewer 2 Report

The manuscript concerns analysis of dietary protein intake in a large number of elderly chinese. The data overall confirm the importance of protein intake for maintenance of muscle mass, though protein distribution across meals did not appear crucial. The manuscript does need improvement as follows:

1. Throughout the researchers use the term non-muscle loss and muscle loss, as this is a cross sectional study they have not looked at loss just muscle mass. They need to change this to normal muscle mass and low muscle mass throughout.

2. The abstract needs revision:

Line 15: add "using" after ASM was obtained.

They must state how normal muscle mass and low muscle mass were defined.

They should add p values for the key significant findings.

They only mention that total protein intake per day was lower in the low muscle mass group and do not mention that if this was expressed as protein/kg/BW/d that the low muscle mass group had higher intake. This must be included.

I think the sentence on line 23 and 24 could be removed.

Line 26: muass should be mass

line 27: shoube change to should be

For the last sentence of the abstract they recommend that protein intake per meal should be increased but they found no evidence for this, I suggest they just advice and increase in protein intake.

3. The U shaped curve for protein intake an risk of low muscle mass should be discussed further, especially the high protein intake and increased risk. This is counterintuitive. They do make some discussion but this needs more on sacopenic obesity.

Author Response

Please check out the uploaded reply, thanks.
